# How Educational Background Influences Recruitment Evaluation: Evidence from Event-Related Potentials

**DOI:** 10.3390/bs15060832

**Published:** 2025-06-19

**Authors:** Bin Ling, Yihan Wang

**Affiliations:** Business School, Hohai University, Nanjing 211100, China; 221313040043@hhu.edu.cn

**Keywords:** internship enterprise, elite university, ERPs, P200, N300

## Abstract

This study used event-related potentials (ERPs) to examine how candidates’ educational background (elite vs. non-elite universities) and prior internship experience (Fortune 500 vs. non-Fortune 500 enterprises) influence recruitment evaluations. Thirty-two participants completed a 2 × 2 within-subjects design task. Behavioral data indicated that applicants with Fortune 500 internships and graduates from elite universities received higher evaluation scores. ERP results revealed that Fortune 500 experience elicited larger P200 amplitudes (reflecting early attention). Crucially, this effect was modulated by educational background as only candidates from elite universities showed both enhanced P200 and reduced N300 amplitudes (suggesting efficient later processing). These findings indicate that recruiters dynamically allocate attention based on academic prestige (P200) and evaluate semantic congruence between education and employer reputation (N300), providing neurophysiological evidence for educational bias in hiring.

## 1. Introduction

In the highly competitive job market, resume screening serves as the first critical checkpoint in corporate recruitment, where its efficiency and criteria directly impact the effectiveness and fairness of human resource management ([21]). As the initial step in the hiring process, resumes not only function as a “stepping stone” for job seekers to showcase their strengths and secure interview opportunities but also represent a crucial mechanism for employers to efficiently identify talent and enhance the equity and efficacy of recruitment practices. This process aligns with Dual-Process Models of Impression Formation ([3]; [13]), which posit that evaluators rely on both rapid, heuristic-based judgments and deliberate, systematic processing when assessing candidates.

The prominence of educational prestige in resume screening exemplifies the heuristic pathway of impression formation. Graduates from elite universities enjoy distinct advantages in recruitment processes ([26]; [28]; [30]), a phenomenon rooted in recruiters’ automatic associations between institutional reputation and perceived competence ([7]; [28]). There exists neurocognitive evidence showing how status markers activate neural reward circuits ([6]) and reduce cognitive conflict signals in evaluative tasks ([19]). Conversely, the critical role of internship experiences reflects the systematic processing pathway, where recruiters engage in effortful analysis of candidates’ practical skills and industry exposure ([20]; [4]). These dual processes create inherent paths in resume evaluation as elite credentials facilitate cognitive efficiency through schema-driven judgments, whilst high-quality internships demand a resource-intensive verification of skill-specific evidence.

Resume screening, functioning as the gatekeeper of talent recruitment processes, holds undeniable significance and has sustained scholarly attention ([10]). Existing research predominantly focuses on behavioral-level analyses, such as empirical studies on resume–job fit ([31]), discriminatory biases in screening practices ([23]), and the impact of resume structure and algorithmic screening technologies ([24]; [32]). A limited number of studies have begun employing eye-tracking techniques to investigate how visual attributes of resumes (e.g., background color, font, photographs) influence recruiters’ attentional patterns and psychological responses ([27]; [29]). However, it cannot capture the dynamic neural computations underlying decision conflicts when elite credentials coexist with suboptimal internships. While these investigations provide valuable insights into resume screening dynamics, critical gaps persist. Specifically, there is a lack of neuroscientific methods to explore the underlying neurocognitive mechanisms, particularly how a recruiters’ brain activity dynamically adapts when evaluating resumes with divergent backgrounds. To address this, our study employs event-related potentials (ERPs) technology, which has a unique capacity to reveal implicit social cognition ([1]) and capture the millisecond-level neural processes underlying recruiters’ automatic responses to candidate credentials—which is precisely what ERPs excel at measuring ([14]).

This study employs two typical ERPs components, P200 and N300, to elucidate the neurocognitive mechanisms underlying the elite university effect. Research demonstrated that the P200 component reflects early attentional resource allocation and information filtering processes ([5]). P200 amplitude increases when tasks demand greater attentional engagement and decreases otherwise ([37]). We thus hypothesize that during resume screening recruiters allocate heightened early attentional resources when evaluating candidates from elite universities and with Fortune 500 internship experience. This neural signature should be detectable through P200 amplitude modulation. Moreover, the N300 component is critically associated with semantic information processing, particularly in contexts of semantic incongruity ([9]; [16]). Studies reveal that semantic conflicts elicit significantly enhanced N300 amplitudes, indicating increased cognitive resource mobilization for conflict resolution ([16]). We hypothesize that when candidates possess both elite academic credentials and Fortune 500 internship experience, the semantic association between ‘prestigious university’ and ‘Fortune 500 internship’ is inherently cohesive (shorter semantic distance). We predict that this configuration would elicit smaller N300 amplitudes. In contrast, when elite university graduates lack Fortune 500 internships (greater semantic distance), we anticipate amplified N300 amplitudes, reflecting heightened neural effort to reconcile this conceptual disparity. Based on these neurocognitive mechanisms, we propose the following hypotheses:

**H1.** 
*Elite university candidates with Fortune 500 internships will elicit significantly enhanced P200 amplitudes compared with other conditions, reflecting prioritized allocation of early attentional resources.*


**H2.** 
*Elite university candidates with Fortune 500 internships (high-semantic-coherence condition) will elicit significantly attenuated N300 amplitudes compared to those with elite university experience and non-Fortune 500 internships (low-semantic-coherence condition).*


This study will simulate authentic resume screening scenarios. From a neuroscientific perspective, it aims to elucidate how recruiters unconsciously reconcile educational prestige and internship pedigree during evaluation processes. This will be achieved by measuring their event-related potentials. Specifically, we aim to investigate the neural interplay between automatic processing (initial preference for elite institutions) and controlled processing (deliberate assessment of internship experiences). By integrating these neural signatures, our framework will provide a novel and in-depth understanding of recruiters’ brain activity in complex decision-making contexts. These findings will extend findings on ERP markers of implicit bias and address critical gaps in understanding the neurocognitive dynamics of resume evaluation. Ultimately, these will enhance both the fairness and efficiency of recruitment processes.

## 2. Materials and Methods

### 2.1. Participants

A priori power analysis was conducted using G*Power (version 3.1) to determine the required sample size for a repeated measures ANOVA with two within-subject factors. The analysis assumed a medium effect size (*f* = 0.25), an alpha level of *α* = 0.05, and a power of 1 − *β* = 0.80. The nonsphericity correction was set to *ϵ* = 0.67, and a moderate correlation (*r* = 0.50) between repeated measures was assumed. Results indicated that a minimum of 31 participants would be required to detect a medium-sized effect.

The study adhered to strict ethical guidelines throughout data collection. All data were anonymized by assigning unique participant codes to separate identifiable information. The experiment recruited a total of 32 participants (13 males), aged between 18 and 24 years (*M* = 21.4, *SD* = 1.43). All participants had normal or corrected-to-normal vision and were right-handed. All individuals were enrolled in the Human Resource Management program and had hands-on experience in practical resume screening tasks. All participants in this study had completed at least three months of internship experience in HR-related roles (e.g., résumé screening, candidate evaluation). Prior to the experiment, written informed consent was obtained from each participant after they were fully informed about the experimental procedures and potential risks. The study protocol was approved by the Institutional Review Board of Business School, Hohai University, and conducted in accordance with the ethical principles of the Declaration of Helsinki. Upon completion of the experiment, all participants were provided with monetary compensation for their time and engagement.

### 2.2. Design

The present experiment employed a 2 (graduate institution: elite university vs. non-elite university) × 2 (internship enterprise: Fortune 500 enterprise vs. non-Fortune 500 enterprise) within-subjects design. The dependent variables included participants’ evaluations of job applicants’ suitability, as well as electrophysiological indicators derived from ERP components.

### 2.3. Stimuli

The experimental stimuli comprised 192 items, including 32 graduate institutions (elite vs. non-elite universities) and 16 internship organizations (Fortune 500 vs. non-Fortune 500 companies). The stimulus materials were derived through the following validation procedures. First, for academic institutions, we selected the top 16 universities from both Double First-Class and non-Double First-Class categories.[note 1] This selection was based on the 2022 Chinese university rankings published by Cuaa.Net, an alumni-based university ranking platform. These constituted the elite and non-elite institutional stimuli. The complete university listings are detailed in the Appendix B. Second, for internship organizations, 8 enterprises were chosen from the Hurun Global 500 list as Fortune 500 corporate stimuli, with an additional 8 enterprises selected from non-Global 500 registries to form the non-Fortune 500 category.

Prior to experimentation, all participants completed a validated questionnaire assessing the degree to which these internship organizations reflected employment capability (5-point Likert scale: 1 = “completely unrepresentative” to 5 = “fully representative”). Statistical analysis revealed significant between-group differences, with Fortune 500 enterprises demonstrating markedly higher perceived employment capability (*M* = 4.54, *SD* = 0.41) compared to non-Fortune 500 counterparts (*M* = 1.77, *SD* = 0.62), *p* < 0.001. The complete enterprise listings are detailed in the Appendix A.

### 2.4. Procedure

The experimental paradigm was programmed using E-Prime 3.0 software. Participants initially completed eight practice trials that replicated the main experimental protocol. This ensured procedural comprehension before formal testing began. During the formal experiment, instructions were first displayed centrally on the monitor. Following participant-initiated advancement via keystroke, a fixation cross (+) appeared at the screen center for 1000 ms, followed by the sequential presentation of stimulus materials. Participants evaluated their willingness to grant subsequent interview opportunities using a 5-point Likert-type scale (1 = “extremely unwilling” to 5 = “extremely willing”) based on contextual information. The experimental structure comprised 6 blocks, each containing 32 trials (total *N* = 192 trials), presented in fully randomized order. A 60 s interval separated successive blocks to mitigate fatigue effects. The complete stimulus presentation sequence and temporal parameters are illustrated in Figure 1.

### 2.5. EEG Recording and Analysis

Electroencephalographic (EEG) activity was recorded using a 64-channel electrode cap (Brain Products GmbH, Gilching, Germany) and an actiCHamp amplifier (Brain Products GmbH, Gilching, Germany). Online recording parameters included a bandpass filter of 0.1–40 Hz, a sampling rate of 500 Hz, and impedance for all electrodes was maintained below 5 kΩ. During online recording, FCz served as the reference electrode, and the ground electrode (GND) was placed on the forehead.

Offline EEG data processing was performed using Brain Vision Analyzer 2.2 software (Brain Products GmbH, Gilching, Germany). Following data import, the reference electrode was re-referenced offline to the average of the bilateral mastoids (M1/M2). The data were then bandpass filtered from 0.1 to 40 Hz to optimize signal quality. Independent component analysis (ICA) was subsequently employed to remove ocular artifacts. EEG epochs were extracted from 200 ms prior to stimulus onset to 1000 ms post-stimulus, and baseline correction was performed using the 200 ms pre-stimulus interval (−200 to 0 ms). Following manual artifact rejection to remove trials contaminated by blinks, muscle activity, and other sources of noise, the remaining artifact-free epochs were averaged separately for each experimental condition to generate grand average waveforms. After artifact rejection, an average of 44.7 (*SD* = 4.5) trials per condition remained. Trials exceeding ±100 μV or containing eye blinks were excluded, with an average rejection rate of 7.4%. Additionally, participants were excluded from analysis if more than 15% of their trials had a response time below 100 ms ([35]). All recruited participants successfully met this predefined criterion.

For the P200 component, we selected the mean amplitude within the 210–280 ms time window. Frontal (F3, Fz, F4) and central (C3, Cz, C4) regions were chosen as the regions of interest (ROIs) for analysis. This selection was based on established evidence that the P200 component peaks between 200 and 300 ms post-stimulus over fronto-central sites during visual attention tasks, reflecting early attentional engagement to motivationally salient stimuli ([22]; [34]). Additionally, the 300–350 ms time window was selected for the analysis of the N300 component, with the same electrode sites serving as the ROIs. This choice was based on the N300’s established role in semantic incongruity processing, with maximal amplitudes typically observed at 250–400 ms over frontal regions ([8]; [11]). All electrode locations adhered to the International 10–20 system, ensuring standardization of spatial analysis.

## 3. Results

### 3.1. Behavioral Results

A repeated measures ANOVA with factors of graduate institution (elite vs. non-elite) and internship enterprise (Fortune 500 vs. Non-Fortune 500) was conducted on the job application ratings. The analysis revealed a significant main effect of internship enterprise, indicating that job applicants with Fortune 500 internship experience received significantly higher ratings (*F*(1, 31) = 52.04, *p* = 0.0003, *η*^2^*p* = 0.63, and 95%CI [4.739, 15.823]). A significant main effect of graduate institution was also found, meaning that graduates of elite universities have obvious advantages in scoring (*F*(1, 31) = 120.46, *p* = 0.0002, *η*^2^*p* = 0.80, and 95%CI [9.383, 22.612]). The interactions between graduate institutions and internship enterprises were marginally significant (*F* (1, 31) = 3.60, *p* = 0.070, *η*^2^*p* = 0.09, and 95%CI [−0.038, 7.383]). Although the effect approached significance (*p* = 0.070), it did not reach the conventional threshold of 0.05.

### 3.2. ERPs Results

#### 3.2.1. P200 (210~280 ms)

A 2 (graduate institution: elite vs. non-elite) × 2 (internship enterprise: Fortune 500 vs. non-Fortune 500) × 2 (brain region: frontal vs. central) repeated-measures ANOVA conducted on electrodes spanning frontal (F3, Fz, F4) and central (C3, Cz, C4) regions revealed significant findings. The main effect of brain region was significant, with greater P200 amplitudes observed in central regions (*M* = 1.00, *SE* = 0.39) compared to frontal regions (*M* = 0.40, *SE* = 0.43), *F*(1, 31) = 8.84, *p* = 0.006, *η*^2^*p* = 0.22, and 95%CI [1.170, 8.351]. The main effect of internship enterprise reached significance, demonstrating larger P200 amplitudes elicited by Fortune 500 corporate information (*M* = 1.19, *SE* = 0.43) relative to that from non-Fortune 500 enterprise information (*M* = 0.21, *SE* = 0.42), *F*(1, 31) = 12.71, *p* = 0.001, *η*^2^*p* = 0.29, and 95%CI [3.181, 9.873]. As hypothesized in H1, a significant interaction emerged between graduate institution and internship enterprise *F*(1, 31) = 4.95, *p* = 0.033, *η*^2^*p* = 0.14, and 95%CI [0.463, 2.048] (see Figure 2). Simple effects analysis revealed that among graduates from elite universities, Fortune 500 internship conditions elicited significantly enhanced P200 amplitudes (*M* = 2.61, *SE* = 0.53) compared to those in non-Fortune 500 conditions (*M* = 0.95, *SE* = 0.54) and *F*(1, 31) = 15.30, *p* = 0.002, *η*^2^*p* = 0.33, and 95%CI [0.574, 2.809]. No significant differences were observed in the non-elite university group (*p* = 0.434). The mean P200 amplitude is detailed in Figure 3, with corresponding topographic mappings presented in Figure 4 According to these results, H1 was supported.

#### 3.2.2. N300 (300~350 ms)

A 2 (graduate institution: elite vs. non-elite) × 2 (internship enterprise: Fortune 500 vs. non-Fortune 500) × 2 (brain region: frontal vs. central) repeated-measures ANOVA conducted on electrodes spanning frontal (F3, Fz, F4) and central (C3, Cz, C4) regions revealed significant findings. The main effect of brain region was significant, with greater N300 amplitudes observed in frontal regions (*M* = −3.24, *SE* = 0.48) compared to central regions (*M* = −1.82, *SE* = 0.49) and *F*(1, 31) = 66.66, *p* = 0.0006, *η*^2^*p* = 0.68, and 95%CI [−4.838, −0.238]. Consistent with H2, the interaction between graduate institution and internship enterprise was significant for N300 amplitude *F*(1, 31) = 9.77, *p* = 0.004, *η*^2^*p* = 0.24, and 95%CI [−6.583, −0.428] (see Figure 5). Simple effects analysis revealed that among graduates from elite universities, Fortune 500 internship conditions elicited significantly reduced N300 amplitudes (*M* = −2.98, *SE* = 0.61) compared to non-Fortune 500 conditions (*M* = −4.65, *SE* = 0.57) *F*(1, 31) = 4.30, *p* = 0.006, *η*^2^*p* = 0.12, and 95%CI [−6.196, −0.265]. No significant differences were observed in the non-elite university group (*p* = 0.163). Neither the main effect of graduate institution nor internship enterprise, nor other interaction effects reached statistical significance (*p* = 0.571). The mean N300 amplitude is detailed in Figure 3, with corresponding topographic mappings presented in Figure 4. According to these results, H2 was supported.

## 4. Discussion

This study revealed that the candidates’ educational background and Fortune 500 internship experiences significantly influenced hiring evaluations during resume screening tasks, demonstrating a pronounced academic prestige bias. Specifically, graduates from elite institutions consistently received higher ratings, consistent with the “elite university effect” cognitive stereotype ([7]; [30]; [32]). A key finding was the synergistic interaction between elite education and Fortune 500 internships: candidates with both attributes received significantly higher evaluations than non-elite counterparts with equivalent internship credentials. These findings suggest that academic prestige acts not only as an initial competence screening threshold but also guides subsequent information processing through cognitive schemas. Recruiters demonstrated selective information weighting based on education, reflecting heuristic mechanisms in decision-making.

The P200 component, recognized as a neurophysiological index of early attentional engagement in motivationally salient stimuli ([5]; [37]), manifested distinct activation patterns that align with value-based attention allocation frameworks. In this study, central-region P200 amplitudes were enhanced, aligning with established central-parietal network functions in early sensory processing ([18]). This suggests heightened sensory engagement during career-related evaluations. Notably, Fortune 500 internship cues elicited larger P200 amplitudes than non-top-tier equivalents, supporting behavioral economics findings on prestige-driven attention capture ([33]). Crucially, elite education amplified this P200 enhancement, revealing neurocognitive mechanisms of early attention modulation by academic prestige. When recruiters’ positive expectations were triggered by elite credentials, the cognitive system showed increased sensitivity to associated information, consistent with prefrontal-parietal gain control mechanisms ([17]). This aligns with the cognitive neuroscience “expectancy–value” framework, where neural responses depend on both incentive value and goal attainment expectancy ([36]). Such neuromodulation may enhance neural efficiency in identifying high-potential candidates.

The N300 component, a neural marker of semantic network dynamics during social information processing, demonstrates sensitivity to semantic incongruence and increased semantic distance during information integration ([9]; [16]). In this study, we observed enhanced N300 amplitudes over frontal electrode clusters. These findings align with the role of frontal cortices in conflict monitoring and cognitive control ([12]), suggesting automated semantic unification during resume evaluation. Notably, Fortune 500 internship information paired with elite educational backgrounds showed reduced N300 amplitudes, indicating lower cognitive load in schema-congruent processing. In contrast, non-Fortune 500 internship records elicited stronger N300 responses in elite academic contexts. This implies that combining non-Fortune 500 companies with elite backgrounds violated expectations, requiring increased neural effort to resolve semantic mismatches. Critically, academic prestige modulated this semantic priming effect. For non-elite candidates, Fortune 500 internships did not evoke significant N300 variations. This may stem from reduced attentional engagement during initial information processing. Such attenuated early processing likely impaired subsequent semantic efficiency, supported by reduced frontal theta synchronization ([15]).

Collectively, the P200 and N300 components represent two distinct neurocognitive stages in resume screening. The P200 corresponds to an attentional orienting phase, during which recruiters selectively focus on academic backgrounds. This is supported by differential activation observed in the temporoparietal junction. The subsequent stage, indexed by the N300, involves semantic integration. Here, prefrontal–striatal circuits assess the congruence between academic prestige and internship quality, ultimately shaping evaluation outcomes. These two stages offer a novel framework for understanding cognitive biases in resume screening. Specifically, educational background not only directs early attention but also impacts later evaluation processes tied to self-concept consistency. This temporal hierarchy provides neurophysiological support for the Dual-Process Models of Impression Formation ([2]; [25]).

This study provides neurophysiological evidence elucidating the dual-stage cognitive mechanisms in hiring decisions, offering empirical support for the Dual-Process Models of Impression Formation. Our findings demonstrate that elite educational backgrounds preferentially capture attentional resources through enhanced P200 amplitudes while modulating semantic integration efficiency via N300 amplitude variations. This temporal dynamic pattern provides tentative support for the hypothesized dissociation and interaction between automatic heuristic processing and controlled evaluation in personnel selection. Notably, we observed a neural-level association pattern between elite degrees and Fortune 500 internship experiences. This finding contributes to understanding how cognitive schemas could shape social evaluations. In addition, the findings offer multidimensional applications for optimizing recruitment practices. First, the revealed “prestige-driven attentional capture” mechanism provides a scientific basis for designing targeted interventions (e.g., attentional retraining programs to reduce automatic responses to elite cues). And for job applicants, the demonstrated cognitive load reduction (N300 attenuation) through education–experience consistency directly guides resume optimization and AI interview algorithm development. These applications collectively promote more equitable hiring practices while maintaining selection efficiency.

## 5. Conclusions, Limitations and Future Directions

This study delineates a neurocognitive trajectory through which graduate institution prestige and internship enterprise status modulate distinct stages of career-related information processing. The P200 and N300 findings collectively reveal a spatiotemporal dynamic where high-prestige stimuli (Fortune 500 enterprises) first capture amplified attentional resources in central regions (indexed by P200), followed by reduced frontal conflict signals (indexed by N300) specifically in elite university graduates. This two-stage mechanism of early salience prioritization and later schema-driven conflict resolution suggests that institutional identity hierarchically gates neural responses. It particularly demonstrates elite graduates’ enhanced attention to prestigious opportunities and reduced cognitive dissonance during self-relevant integration.

Limitations and future directions: While this study offers novel insights into the neurocognitive mechanisms underlying hiring biases, several limitations should be acknowledged. First, although the laboratory-based experimental design ensures methodological control, it may lack ecological validity when compared to real-world recruitment contexts. Future studies could address this by conducting field experiments in actual hiring settings, such as partnering with corporate recruitment teams to observe decision-making processes in real time. Second, as our sample consisted exclusively of HR management students (albeit with internship experience) the findings have inherent limitations in generalizability to actual recruitment contexts. We therefore strongly recommend that future research prioritize replication with practicing HR professionals as the primary validation step. Subsequent extensions could then examine the following: (a) expert–novice comparisons in evaluator cognition, (b) cross-cultural cohorts to assess sociocultural modulation effects, and (c) age-diverse populations to explore developmental aspects of bias formation. Third, while our standardized stimuli controlled for extraneous variables, the repetitive presentation of stylized candidate profiles introduces potential familiarity effects. Such learned schematic processing may systematically bias participants’ responses. Future research should consider employing novel or less conventional stimuli.

## Figures and Tables

**Figure 1 behavsci-15-00832-f001:**
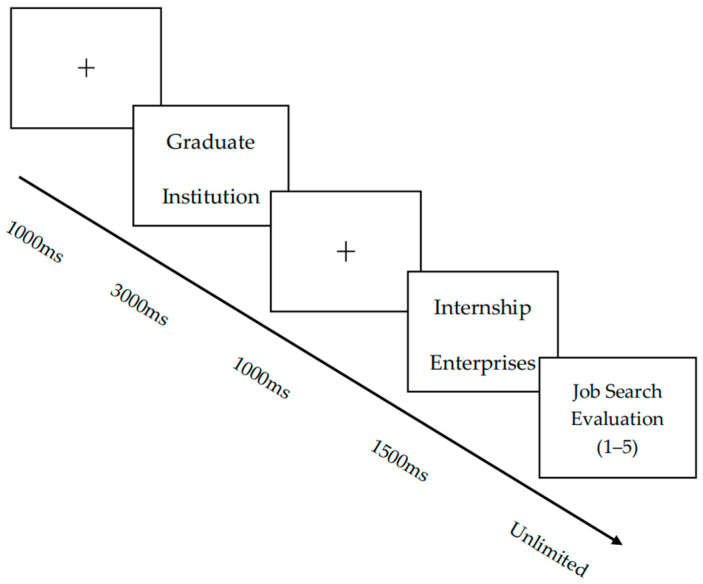
Experiment procedure.

**Figure 2 behavsci-15-00832-f002:**
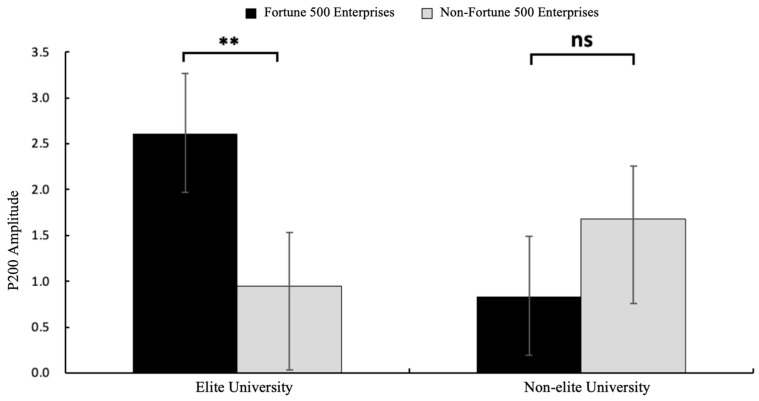
The results of P200 amplitude (μV) in each condition. Error bars represent standard errors. (ns: nonsignificant, ** *p* < 0.01).

**Figure 3 behavsci-15-00832-f003:**
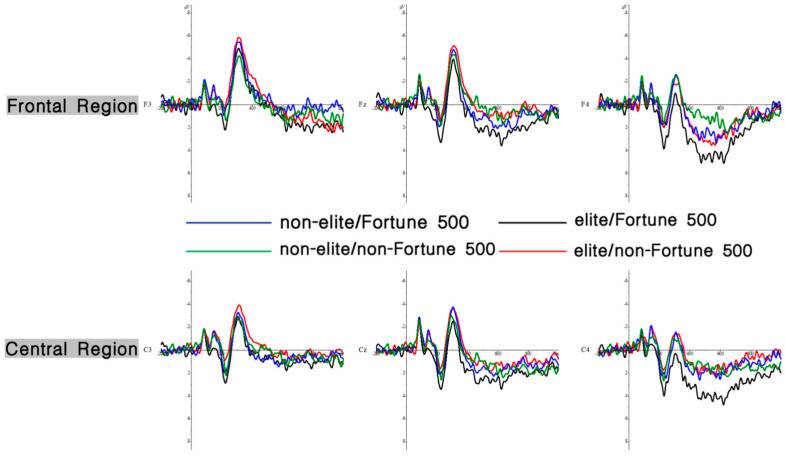
ERPs results. The total average waveforms at six representative electrodes in the midzone (FZ, F3, F4, CZ, C3, and C4) are presented. Positive voltage is plotted downward. The *x*-axis represents time (ms) relative to stimulus onset (0 ms), and the *y*-axis represents amplitude values (μV).

**Figure 4 behavsci-15-00832-f004:**
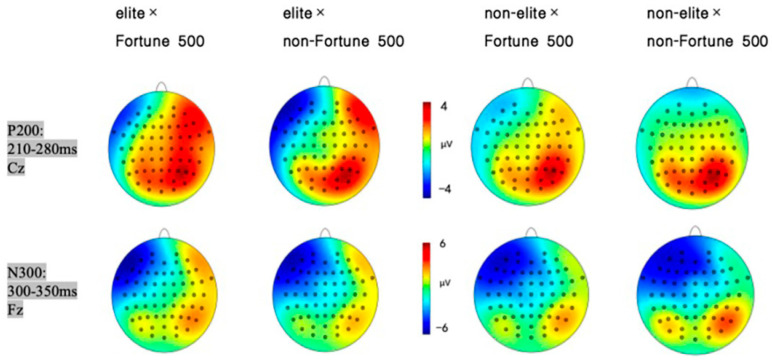
P200 (Cz, 210–280 ms) and N300 (Fz, 300–350 ms) scalp distributions across elite/non-elite × Fortune/non-Fortune 500 conditions.

**Figure 5 behavsci-15-00832-f005:**
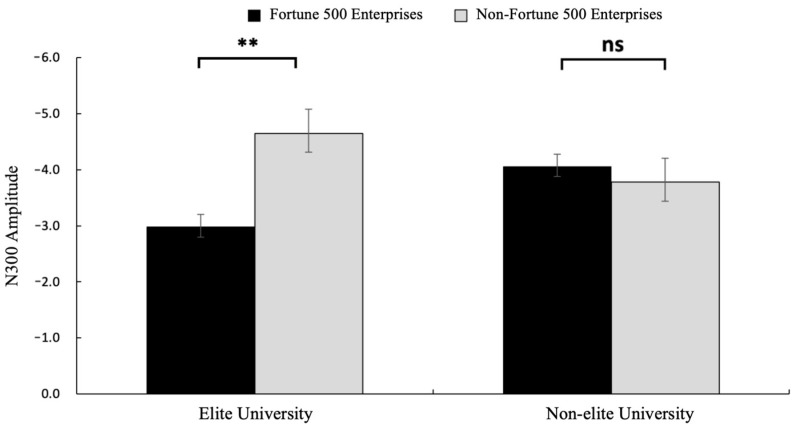
The results of N300 amplitude (μV) in each condition. Error bars represent standard errors (ns: nonsignificant, ** *p* < 0.01).

## Data Availability

The original contributions presented in this study are included in the article. Further inquiries can be directed to the corresponding author.

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
