# Peer review of "How Educational Background Influences Recruitment Evaluation: Evidence from Event-Related Potentials"

_behavsci, 2025, doi:10.3390/bs15060832_

Round 1
Reviewer 1 Report
Comments and Suggestions for Authors
The study asks how two résumé cues (graduation from an elite versus non-elite Chinese university and internship experience at a Fortune 500 versus non-Fortune 500 firm) jointly shape recruiters’ assessments of applicants, and what neurocognitive stages mediate this process during résumé screening. Most prior hiring-bias work stops at behavioural ratings or, more recently, eye-tracking; almost none probes the temporal dynamics of neural processing during résumé evaluation.
That said, the interaction of educational pedigree and internship quality has been discussed behaviourally; the contribution here lies less in the behavioural finding than in mapping it onto time-resolved neural indices.
All 32 participants are HRM undergraduates aged 18–24: their limited professional experience and cultural homogeneity restrict generalisability! I think that a replication with practising recruiters and with participants outside mainland China would make the claims more robust.
I even think that providing the full 32-university and 16-enterprise sets, in English and Chinese, could be useful for readers that can assess semantic familiarity and prestige confounds.
The data show an interaction between academic prestige and internship status on both ratings and ERPs but the manuscript leaps from a student-lab finding to policy claims about “urgent need to eliminate academic discrimination”.
Your manuscript would benefit from several clarifications and extensions. First, present the study’s hypotheses in a formal, numbered list and link each one explicitly to the corresponding statistical test; this will help readers follow the logic from prediction to result.
Next, acknowledge the limits of external validity that arise from recruiting only undergraduate HRM students aged 18–24 in mainland China, and consider collecting confirmatory data from professional recruiters to bolster generalizability.
In the theoretical framing, move beyond the label of “academic prestige bias” by linking your interpretation to established dual-process models of impression formation or expectancy-violation theory and balance the emphasis on fairness implications with practical applications, such as training programs that could mitigate prestige-driven attentional capture, and temper policy claims by acknowledging the laboratory, hypothetical-résumé setting.
Comments on the Quality of English LanguageEnglish is serviceable and professional language edit would sharpen clarity and polish.
Author Response
Thank you very much for your helpful suggestions. Our paper has benefited substantially from your insightful comments, resulting in a much-improved paper in many respects. In the following, we provide our point-by-point responses (in regular font) to your original comments (reproduced in bold and italics). The quotations from the manuscript is marked as italics and red color.
- The study asks how two résumé cues (graduation from an elite versus non-elite Chinese university and internship experience at a Fortune 500 versus non-Fortune 500 firm) jointly shape recruiters’ assessments of applicants, and what neurocognitive stages mediate this process during résumé Most prior hiring-bias work stops at behavioural ratings or, more recently, eye-tracking; almost none probes the temporal dynamics of neural processing during résumé evaluation. That said, the interaction of educational pedigree and internship quality has been discussed behaviourally; the contribution here lies less in the behavioural finding than in mapping it onto time-resolved neural indices.
Thank you for highlighting this important point. We fully agree that the interaction between educational pedigree and internship experience has been previously explored in behavioral research. However, as the reviewer rightly noted, our study advances this line of inquiry by investigating the neurocognitive mechanisms underlying recruiters’ evaluations, thereby addressing a critical gap in the literature.
Specifically, while prior studies have largely focused on behavioral outcomes or eye-tracking metrics, our research employs event-related potentials (ERPs) to capture the temporal dynamics of how résumé cues are processed in real-time. This allows us to uncover distinct stages of neural processing—such as early attentional allocation (indexed by P2) and later evaluative processes (indexed by N3)—that help explain how and when recruiters integrate educational and internship cues during decision-making.
In this sense, the core contribution of our study lies not only in replicating behavioral effects but in mapping these effects onto temporally precise neural indices, thus offering deeper insight into the cognitive architecture of hiring judgments. We have revised the manuscript to make this contribution more explicit (see page 10, lines 328-344).
- All 32 participants are HRM undergraduates aged 18–24: their limited professional experience and cultural homogeneity restrict generalisability! I think that a replication with practising recruiters and with participants outside mainland China would make the claims more robust.
We fully acknowledge that the sample composition imposes limitations on the generalizability of our findings across professional and cultural contexts. The reviewer’ s critique is valid, and we agree that broader participant pools would strengthen the robustness of the conclusions.
Despite the inherent limitations in generalizability, the current sample was selected based on specific research objectives. By choosing participants from the same age group, academic discipline, and cultural region, we deliberately controlled for potential confounding variables. This approach enables a clearer examination and identification of the relationships among research variables within specific educational and cultural frameworks, while minimizing interference stemming from differences in professional backgrounds, career experiences, or multicultural diversity. Consequently, it facilitates the establishment of internal validity. While we regret that additional data collection with practicing professionals is currently infeasible due to time constraints, we would like to clarify that all participants in this study had completed at least three months of internship experience in HR-related roles (e.g., résumé screening, candidate evaluation) prior to the experiment.
While our findings are context-bound, they offer two key contributions: (1) Mechanistic Insights: The results reveal how early-career individuals interpret recruitment cues, which is critical for designing training programs for future HR professionals. (2) Baseline Data: This study establishes a benchmark for future cross-cultural comparisons, as suggested by the reviewer.
We fully concur with your suggestion and agree that future research should indeed expand the breadth and depth of the sample to enhance the generalizability of the findings. We plan to pursue the following research directions in future studies: (1) Including working recruiters or human resources professionals as participants; (2) Extending cultural backgrounds and geographic regions to conduct cross-cultural studies. These will strengthen the study's generalizability and cross-cultural validity.
‘All individuals were enrolled in the Human Resource Management program and possessed hands-on experience in practical resume screening tasks. All participants in this study had completed at least three months of internship experience in HR-related roles (e.g., résumé screening, candidate evaluation).’(see page 3 lines 118-121 )
- I even think that providing the full 32-university and 16-enterprise sets, in English and Chinese, could be useful for readers that can assess semantic familiarity and prestige confounds.
Thank you for this insightful suggestion. We have added the complete lists of 32-university and 16-enterprise on page 11 to 12 (Appendix A and Appendix B),with bilingual names.
4.The data show an interaction between academic prestige and internship status on both ratings and ERPs but the manuscript leaps from a student-lab finding to policy claims about “urgent need to eliminate academic discrimination”.
We sincerely appreciate your critical observation regarding the overinterpretation of our findings in the original manuscript. We apologize for this overreach and agree that extrapolating student-lab results to broad policy prescriptions was unwarranted. In the revised manuscript, we have removed all claims about an “urgent need to eliminate academic discrimination” and other policy-centric language. These statements have been replaced with conclusions strictly grounded in our experimental data, as detailed from lines 345 to 355.
‘The study delineates a neurocognitive trajectory through which academic prestige and internship enterprise status modulate distinct stages of career-related information processing. The P200 and N300 findings collectively reveal a spatiotemporal dynamic: high-prestige stimuli (Fortune 500 enterprises) first capture amplified attentional resources in central regions (indexed by P200), followed by reduced frontal conflict signals (indexed by N300) specifically in elite university conditions. This two-stage mechanism, early salience prioritization and later schema-driven conflict resolution, suggests that institutional identity hierarchically gates neural responses, with elite graduates exhibiting both enhanced motivated attention to prestigious opportunities and diminished cognitive dissonance when integrating these stimuli into self-relevant frameworks.’
- Your manuscript would benefit from several clarifications and extensions. First, present the study’ s hypotheses in a formal, numbered list and link each one explicitly to the corresponding statistical test; this will help readers follow the logic from prediction to result.
Thank you for this valuable suggestion. We have carefully revised the manuscript to improve the clarity and traceability of our hypotheses. Specifically:
- In the Introduction, we now present the hypotheses as a numbered list (H1, H2) in a dedicated paragraph before the Methods section (lines 88-94). Each hypothesis explicitly references the predicted neural markers (P200 and N300) and their cognitive interpretations.
‘Based on these neurocognitive mechanisms, we propose the following hypotheses:
H1: Elite university candidates with Fortune 500 internships will elicit significantly enhanced P200 amplitudes compared with other conditions, reflecting prioritized allocation of early attentional resources.
H2: Elite university candidates with Fortune 500 internships (high semantic coherence condition) will elicit significantly attenuated N300 amplitudes compared to those with elite university and non-Fortune 500 internship (low semantic coherence condition).’
- In the Results, we clearly explain whether the hypothesis has been verified, as detailed on 6 and p.7, lines 226-234 and lines 244-254.
‘As hypothesized in H1, ... According to the results above, H1 got supported.’ and ‘Consistent with H2, ... According to the results above, H2 got supported.’
- Next, acknowledge the limits of external validity that arise from recruiting only undergraduate HRM students aged 18–24 in mainland China, and consider collecting confirmatory data from professional recruiters to bolster generalizability.
Thank you for raising these critical points regarding the generalizability of our findings and the potential value of sharing stimulus materials. We acknowledge the limitations of our participant sample and fully agree that future replications with experienced recruiters and cross-cultural samples would strengthen the robustness of the conclusions. We have explicitly addressed this limitation in the revised manuscript (lines 356-368) and emphasized the need for future validation in practitioner populations.
‘Limitations and Future Directions: While this study offers novel insights into the neurocognitive mechanisms underlying hiring biases, several limitations should be acknowledged. First, although the laboratory-based experimental design ensures methodological control, it may lack ecological validity when compared to real-world recruitment contexts. Future studies could address this by conducting field experiments in actual hiring settings, such as partnering with corporate recruitment teams to observe decision-making processes in real time. Second, our sample consisted exclusively of Chinese university students with internship experience, which limits the generalizability of the findings to working professionals and more diverse demographic groups. To enhance external validity, subsequent research should incorporate (a) experienced HR professionals to compare expert vs. novice evaluator cognition, (b) cross-cultural cohorts to examine how sociocultural norms modulate neural responses, and (c) age-diverse working populations to assess developmental differences in bias formation.’
- In the theoretical framing, move beyond the label of “academic prestige bias” by linking your interpretation to established dual-process models of impression formation or expectancy-violation theory and balance the emphasis on fairness implications with practical applications, such as training programs that could mitigate prestige-driven attentional capture, and temper policy claims by acknowledging the laboratory, hypothetical-résumé setting.
Thank you for this suggestion. These revisions are organized as followed:
- Our revision about linking interpretation to established dual-process models of impression formation or expectancy-violation theory:
‘The P200 reflects an attentional orienting phase where recruiters allocate selective attention based on academic background, as evidenced by differential activation in the temporo-parietal junction. The subsequent semantic integration stage indexed by N300 reveals how prefrontal-striatal circuits evaluate congruence between educational institutions and internship quality and ultimately influences evaluation outcomes. These two stages provide a novel perspective for understanding cognitive biases in resume screening, emphasizing that educational background not only shapes early attention but also influences later evaluation processes related to self-concept consistency. This temporal hierarchical structure provides neurophysiological evidence for Dual-Process Models of Impression Formation (Amodio, 2019; Monroe et al., 2018).’(see lines 318-327)
Reference
Amodio, D. M. (2019). Social Cognition 2.0: An Interactive Memory Systems Account. Trends in Cognitive Sciences, 23(1), 21–33. https://doi.org/10.1016/j.tics.2018.10.002
Monroe, B. M., Koenig, B. L., Wan, K. S., Laine, T., Gupta, S., & Ortony, A. (2018). Re-examining dominance of categories in im-pression formation: A test of dual-process models. Journal of Personality and Social Psychology, 115(1), 1–30. https://doi.org/10.1037/pspa0000119
- Our revision about balancing the emphasis on fairness implications with practical applications are as follows:
‘Besides, the findings offer multidimensional applications for optimizing recruitment practices. First, the revealed "prestige-driven attentional capture" mechanism provides a scientific basis for designing targeted interventions (e.g., attentional retraining programs to reduce automatic responses to elite cues).’(see lines 336-340)
As you suggested, we have replaced policy claims in original “Conclusion” section by results-related conclusions.
Reviewer 2 Report
Comments and Suggestions for Authors
thank you for the opportunity to review the manuscript.
here are my comments;
- the abstract should also mention the methodology of the research - maintain the 200 word limit
- the introduction should better justify the conduct of the study while highlighting its potential contributions, gaps in the literature being addressed, and contextual setting of the study.
- sample size is accurately calculated with G*power
- while the experiment method seems appropriate, authors should clearly explain their approach, why it was used and provide support from literature (similar studies)
- authors should explain the ethical means of conduct for collecting data (e.g. consent form, anonymity etc)
- analysis is correctly presented
- in the discussion section, authors must clearly outline the contributions of the research while comparing results with the existing (recent) studies in the literature.
- conclusions seems contrived and do not reflect the actual interpretation of the results. authors must revise this section.
- perhaps including a theoretical setting in the study can increase its merits
- limitations and/or recommendations?
Comments on the Quality of English Language
- authors should maintain an integrated tone throughout the study.
- some lexical, grammatical, and sentence structure errors can be seen throughout the text which should be resolved.
Author Response
Thank you very much for your constructive feedback, which has enabled us to improve our paper significantly. We have thoroughly revised the paper and, as a result, the revised papers has greatly benefited from all of your thoughtful suggestions. Below, we provide our point-by-point response (in regular font) to your original comments (in bold and italics). The quotations from the manuscript is marked as italics and red color.
- The abstract should also mention the methodology of the research - maintain the 200 word limit.
Thank you very much. As you suggested, we have revised the Abstract to include key methodological details while maintaining the 200-word limit, as follow:
‘We adopted a 2 (graduate institution: elite universities vs. non-elite universities) × 2 (internship enterprise: Fortune 500 enterprises vs. non-Fortune 500 enterprise) with-in-subjects design with 32 participants.’
- The introduction should better justify the conduct of the study while highlighting its potential contributions, gaps in the literature being addressed, and contextual setting of the study.
Thank you for your valuable suggestions to our study. We have revised the Introduction on page 1- 3, as follows:
- Revision about contextual setting of the study:
‘In the highly competitive job market, resume screening serves as the first critical checkpoint in corporate recruitment, where its efficiency and criteria directly impact the effectiveness and fairness of human resource management (Ling et al., 2024). As the initial step in the hiring process, resumes not only function as a “stepping stone” for job seekers to showcase their strengths and secure interview opportunities but also represent a crucial mechanism for employers to efficiently identify talent and enhance the equity and efficacy of recruitment practices. This process aligns with Dual-Process Models of Impression Formation (Bellucci, 2023; Freeman & Ambady, 2011), which posit that evaluators rely on both rapid, heuristic-based judgments and deliberate, systematic processing when assessing candidates.
The prominence of educational prestige in resume screening exemplifies the heuristic pathway of impression formation. Graduates from elite universities enjoy distinct advantages in recruitment processes (Moore et al., 2023; Pinto et al., 2024; Ramamurthy & Sedgley, 2019), a phenomenon rooted in recruiters’ automatic associations between institutional reputation and perceived competence (Campbell et al., 2019; Pinto et al., 2024). There exists neurocognitive evidence showing how status markers activate neural reward circuits (Bowyer et al., 2021) and reduce cognitive conflict signals in evaluative tasks (Jia et al., 2021). Conversely, the critical role of internship experiences reflects the systematic processing pathway, where recruiters engage in effortful analysis of candidates’ practical skills and industry exposure (Li & Lu, 2023; Benati et al., 2023). These dual processes create inherent path in resume evaluation, while elite credentials facilitate cognitive efficiency through schema-driven judgments, high-quality internships demand resource-intensive verification of skill-specific evidence.’(see lines 29-51)
- Revision about gaps in the literature being addressed:
‘Resume screening, functioning as the gatekeeper of talent recruitment processes, holds undeniable significance and has sustained scholarly attention (Derous & Ryan, 2019). Existing research predominantly focuses on behavioral-level analyses, such as empirical studies on resume-job fit (Rivera, 2020), discriminatory biases in screening practices (Lu & Li, 2021), and the impact of resume structure and algorithmic screening technologies (Luo & Yu, 2024; Sachs et al., 2024). A limited number of studies have begun employing eye-tracking techniques to investigate how visual attributes of resumes (e.g., background color, font, photographs) influence recruiters’attentional patterns and psychological responses (Pina et al., 2023; Pisanelli, 2022), it cannot capture the dynamic neural computations underlying decision conflicts when elite credentials coexist with suboptimal internships. While these investigations provide valuable insights into resume screening dynamics, critical gaps persist in exploring the underlying neurocognitive mechanisms, particularly the lack of neuroscientific methods to uncover how recruiters’ brain activity dynamically adapts when evaluating resumes with divergent backgrounds. To address this, our study employs Event-Related Potentials (ERPs) technology, which has unique capacity to reveal implicit social cognition (Addante et al., 2023) and capture the millisecond-level neural processes underlying recruiters' automatic responses to candidate credentials-precisely what ERPs excel at measuring (Ghani et al., 2020).’(see lines 52-69)
- Revision about potential contributions:
‘This study will simulate authentic resume screening scenarios to elucidate, from a neuroscientific perspective, how recruiters unconsciously reconcile educational prestige and internship pedigree during evaluation processes by measuring their event-related potentials. Specifically, we aim to investigate the neural interplay between automatic processing (initial preference for elite institutions) and controlled processing (deliberate assessment of internship experiences). By integrating these neural signatures, our framework will provide a novel and in-depth understanding of recruiters' brain activity in complex decision-making contexts. These findings will extend findings on ERP markers of implicit bias, addresses critical gaps in understanding the neurocognitive dynamics of resume evaluation, ultimately enhancing both the fairness and efficiency of recruitment processes.’(see lines 95-105)
Reference
Bellucci, G. (2023). The organizational principles of impression formation. Cognition, 239, 105550. https://doi.org/10.1016/j.cognition.2023.105550
Freeman, J. B., & Ambady, N. (2011). A dynamic interactive theory of person construal. Psychological Review, 118(2), 247–279. https://doi.org/10.1037/a0022327
Bowyer, C., Brush, C. J., Threadgill, H., Harmon-Jones, E., Treadway, M., Patrick, C. J., & Hajcak, G. (2021). The effort-doors task: Examining the temporal dynamics of effort-based reward processing using ERPs. NeuroImage, 228, 117656. https://doi.org/10.1016/j.neuroimage.2020.117656
Jia, Y., Cui, L., Pollmann, S., & Wei, P. (2021). The interactive effects of reward expectation and emotional interference on cognitive conflict control: An ERP study. Physiology & Behavior, 234, 113369. https://doi.org/10.1016/j.physbeh.2021.113369
Addante, R. J., Lopez‐Calderon, J., Allen, N., Luck, C., Muller, A., Sirianni, L., Inman, C. S., & Drane, D. L. (2023). An ERP measure of non‐conscious memory reveals dissociable implicit processes in human recognition using an open‐source au-tomated analytic pipeline. Psychophysiology, 60(10), e14334. https://doi.org/10.1111/psyp.14334
Ghani, U., Signal, N., Niazi, I. K., & Taylor, D. (2020). ERP based measures of cognitive workload: A review. Neuroscience & Bi-obehavioral Reviews, 118, 18–26. https://doi.org/10.1016/j.neubiorev.2020.07.020
- Sample size is accurately calculated with G*power.
Thank you very much. As you suggested, we have added the process of calculating the sample size through G*Power to 2.1 part, as detailed from lines 108-113:
‘A priori power analysis was conducted using G*Power (version 3.1) to determine the required sample size for a repeated measures ANOVA with two within-subject factors. The analysis assumed a medium effect size (?=0.25), an alpha level of ? =.05, and a power of 1 −? =.80. The nonsphericity correction was set to ? =.67 , and a moderate correlation (? =.50) between repeated measures was assumed. Results indicated that a minimum of 31 participants would be required to detect a medium-sized effect.’
- While the experiment method seems appropriate, authors should clearly explain their approach, why it was used and provide support from literature (similar studies).
Thank you for your constructive feedback. We have revised the Introduction section (lines 57-69) to provide a more detailed justification for our experimental approach, highlighting ERP’s validity for measuring implicit bias in social evaluation tasks and the temporal precision of ERP for capturing rapid neural responses during resume screening:
‘A limited number of studies have begun employing eye-tracking techniques to investigate how visual attributes of resumes (e.g., background color, font, photographs) influence recruiters’ attentional patterns and psychological responses (Pina et al., 2023; Pisanelli, 2022), it cannot capture the dynamic neural computations underlying decision conflicts when elite credentials coexist with suboptimal internships. While these investigations provide valuable insights into resume screening dynamics, critical gaps persist in exploring the underlying neurocognitive mechanisms, particularly the lack of neuroscientific methods to uncover how recruiters’ brain activity dynamically adapts when evaluating resumes with divergent backgrounds. To address this, our study employs Event-Related Potentials (ERPs) technology, which has unique capacity to reveal implicit social cognition (Addante et al., 2023) and capture the millisecond-level neural processes underlying recruiters' automatic responses to candidate credentials-precisely what ERPs excel at measuring (Ghani et al., 2020).’
Reference
Addante, R. J., Lopez‐Calderon, J., Allen, N., Luck, C., Muller, A., Sirianni, L., Inman, C. S., & Drane, D. L. (2023). An ERP measure of non‐conscious memory reveals dissociable implicit processes in human recognition using an open‐source au-tomated analytic pipeline. Psychophysiology, 60(10), e14334. https://doi.org/10.1111/psyp.14334
Ghani, U., Signal, N., Niazi, I. K., & Taylor, D. (2020). ERP based measures of cognitive workload: A review. Neuroscience & Bi-obehavioral Reviews, 118, 18–26. https://doi.org/10.1016/j.neubiorev.2020.07.020
- Authors should explain the ethical means of conduct for collecting data (e.g. consent form, anonymity etc) .
‘The study adhered to strict ethical guidelines throughout data collection. All data were anonymized, we assigned them unique participant codes to separate identifiable information.’
- Analysis is correctly presented.
Thank you for your positive feedback regarding our analytical approach. We appreciate your recognition of our efforts to maintain rigorous statistical standards in this study. To further strengthen the manuscript, we have ensured all statistical tests are explicitly linked to their corresponding hypotheses in the Results section and we verified that all assumptions of the statistical tests were properly met.
- In the discussion section, authors must clearly outline the contributions of the research while comparing results with the existing (recent) studies in the literature.
Thank you for your valuable suggestion. We have carefully revised the Discussion section to:
- Explicitly outline the study’s contributions in a dedicated paragraph (lines 328–344), including:
‘This study provides neurophysiological evidence elucidating the dual-stage cognitive mechanisms in hiring decisions, offering empirical support for Dual-Process Models of Impression Formation. Our findings demonstrate that elite educational backgrounds preferentially capture attentional resources through enhanced P200 amplitudes while modulating semantic integration efficiency via N300 amplitude variations. This temporal dynamic pattern confirms the dissociation and interaction between automatic heuristic processing and controlled evaluation in personnel selection. Notably, we identified a neural-level "halo synergy effect" between elite degrees and Fortune 500 internship experiences, advancing understanding of how cognitive schemas shape social evaluations. Besides, the findings offer multidimensional applications for optimizing recruitment practices. First, the revealed "prestige-driven attentional capture" mechanism provides a scientific basis for designing targeted interventions (e.g., attentional retraining programs to reduce automatic responses to elite cues). And for job applicants, the demonstrated cognitive load reduction (N300 attenuation) through education-experience consistency directly guides resume optimization and AI interview algorithm development. These applications collectively promote more equitable hiring practices while maintaining selection efficiency.’
- Strengthen comparisons with recent literature are detailed on page 9 under the “Discussion” section, including:
‘In the present study, the P200 component was successfully elicited with enhanced amplitude over central regions, aligning with prior ERP studies implicating the central-parietal network in early sensory processing and stimulus salience detection (Irak et al., 2019), reflecting heightened sensory processing in central regions when participants evaluate career-related information.’(see lines 280-284)
‘Notably, internship information from Fortune 500 corporations elicited significantly greater P200 amplitudes compared to non-top-tier enterprises, this finding corroborated behavioral economics research demonstrating that prestigious brands or employers automatically capture attention due to their association with social rewards (Tang & Gray, 2021),’(see lines 284-288)
‘This aligned with the “expectancy value” framework in cognitive neuroscience, where neural response to incentives depend on both perceived value and subjective expectancy of goal attainment (Zeineldin et al., 2022).’(see lines 294-296)
‘The current investigation revealed enhanced N300 amplitudes over frontal electrode clusters, aligned with the established role of frontal cortices in conflict monitoring and higher-order cognitive control (Fraga et al., 2018),’(see lines 301-303)
‘Conversely, non-Fortune 500 internship records elicited heightened N300 responses in elite academic contexts, this indicated that the combination of non-Fortune 500 companies and elite academic backgrounds violated the expectations,’(see lines 307-309)
Reference
Irak, M., Soylu, C., Turan, G., & Çapan, D. (2019). Neurobiological basis of feeling of knowing in episodic memory. Cognitive Neu-rodynamics, 13(3), 239–256. https://doi.org/10.1007/s11571-019-09520-5
Tang, S., & Gray, K. (2021). Feeling empathy for organizations: Moral consequences, mechanisms, and the power of framing. Jour-nal of Experimental Social Psychology, 96, 104147. https://doi.org/10.1016/j.jesp.2021.104147
Zeineldin, M., Patel, A. G., & Dyer, M. A. (2022). Neuroblastoma: When differentiation goes awry. Neuron, 110(18), 2916–2928. https://doi.org/10.1016/j.neuron.2022.07.012
Fraga, F. J., Mamani, G. Q., Johns, E., Tavares, G., Falk, T. H., & Phillips, N. A. (2018). Early diagnosis of mild cognitive impair-ment and Alzheimer’s with event-related potentials and event-related desynchronization in N-back working memory tasks. Computer Methods and Programs in Biomedicine, 164, 1–13. https://doi.org/10.1016/j.cmpb.2018.06.011
- Conclusions seems contrived and do not reflect the actual interpretation of the results. authors must revise this section.
Thank you for your constructive feedback. We have rewritten the Conclusions section to ensure it strictly aligns with the empirical results and focuses on the core neurocognitive findings, as detailed in lines 345-355:
‘The study delineates a neurocognitive trajectory through which graduate institution prestige and internship enterprise status modulate distinct stages of career-related information processing. The P200 and N300 findings collectively reveal a spatiotemporal dynamic: high-prestige stimuli (Fortune 500 enterprises) first capture amplified attentional resources in central regions (indexed by P200), followed by reduced frontal conflict signals (indexed by N300) specifically in elite university graduates. This two-stage mechanism, early salience prioritization and later schema-driven conflict resolution, suggests that institutional identity hierarchically gates neural responses, with elite graduates exhibiting both enhanced motivated attention to prestigious opportunities and diminished cognitive dissonance when integrating these stimuli into self-relevant frameworks.’
- Perhaps including a theoretical setting in the study can increase its merits.
Thank you for your valuable suggestion to strengthen the theoretical foundation of our study. We have revised the Introduction to incorporate a dedicated theoretical framework section (lines 29-51), as follows:
‘In the highly competitive job market, resume screening serves as the first critical checkpoint in corporate recruitment, where its efficiency and criteria directly impact the effectiveness and fairness of human resource management (Ling et al., 2024). As the initial step in the hiring process, resumes not only function as a “stepping stone” for job seekers to showcase their strengths and secure interview opportunities but also represent a crucial mechanism for employers to efficiently identify talent and enhance the equity and efficacy of recruitment practices. This process aligns with Dual-Process Models of Impression Formation (Bellucci, 2023; Freeman & Ambady, 2011), which posit that evaluators rely on both rapid, heuristic-based judgments and deliberate, systematic processing when assessing candidates.
The prominence of educational prestige in resume screening exemplifies the heuristic pathway of impression formation. Graduates from elite universities enjoy distinct advantages in recruitment processes (Moore et al., 2023; Pinto et al., 2024; Ramamurthy & Sedgley, 2019), a phenomenon rooted in recruiters’ automatic associations between institutional reputation and perceived competence (Campbell et al., 2019; Pinto et al., 2024). There exists neurocognitive evidence showing how status markers activate neural reward circuits (Bowyer et al., 2021) and reduce cognitive conflict signals in evaluative tasks (Jia et al., 2021). Conversely, the critical role of internship experiences reflects the systematic processing pathway, where recruiters engage in effortful analysis of candidates’ practical skills and industry exposure (Li & Lu, 2023; Benati et al., 2023). These dual processes create inherent path in resume evaluation, while elite credentials facilitate cognitive efficiency through schema-driven judgments, high-quality internships demand resource-intensive verification of skill-specific evidence.’
Reference
Bellucci, G. (2023). The organizational principles of impression formation. Cognition, 239, 105550. https://doi.org/10.1016/j.cognition.2023.105550
Freeman, J. B., & Ambady, N. (2011). A dynamic interactive theory of person construal. Psychological Review, 118(2), 247–279. https://doi.org/10.1037/a0022327
Bowyer, C., Brush, C. J., Threadgill, H., Harmon-Jones, E., Treadway, M., Patrick, C. J., & Hajcak, G. (2021). The effort-doors task: Examining the temporal dynamics of effort-based reward processing using ERPs. NeuroImage, 228, 117656. https://doi.org/10.1016/j.neuroimage.2020.117656
Jia, Y., Cui, L., Pollmann, S., & Wei, P. (2021). The interactive effects of reward expectation and emotional interference on cognitive conflict control: An ERP study. Physiology & Behavior, 234, 113369. https://doi.org/10.1016/j.physbeh.2021.113369
- Limitations and/or recommendations?
Thank you very much for your valuable comments. As you suggested, we have added limitations and future directions in “Conclusion” section, as detailed from lines 356 to 368:
‘Limitations and Future Directions: While this study offers novel insights into the neurocognitive mechanisms underlying hiring biases, several limitations should be acknowledged. First, although the laboratory-based experimental design ensures methodological control, it may lack ecological validity when compared to real-world recruitment contexts. Future studies could address this by conducting field experiments in actual hiring settings, such as partnering with corporate recruitment teams to observe decision-making processes in real time. Second, our sample consisted exclusively of Chinese university students with internship experience, which limits the generalizability of the findings to working professionals and more diverse demographic groups. To enhance external validity, subsequent research should incorporate (a) experienced HR professionals to compare expert vs. novice evaluator cognition, (b) cross-cultural cohorts to examine how sociocultural norms modulate neural responses, and (c) age-diverse working populations to assess developmental differences in bias formation.’
Reviewer 3 Report
Comments and Suggestions for Authors
Dear authors,
Thank you for the opportunity of reading the results of your research. The topic of the study is interesting and relevant for the actual economic and educational concerns providing solid practical inputs to a better employee selection.
The theoretical background of the study is well references but could be improved by approaching some social theories such as Self-Efficacy Theory, Self-determination theory, Multiple Intelligence Theory of other learning/Social theories. It is important to see if the academical environment and final evaluation are truly the main factors that contributed to the results of 500 Forbes members and not other factors and how social and learning theories explain the results.
Statistical methods should be more clear connected with the theoretical background, better hypotheses formulation would help the reader understand what the results represent according to previous theory and statistics research on the topic.
The results should compare more studies of the impact of the ,,stereotypical cognitive pattern termed the “elite university effect” on the success of top fortunes. This argument is not very well sustained by the study discussions and conclusions.
Practical contributions are very clear stated, but the study must underline a more solid scientific and theoretical background, also connect the obtained results with previous similar results.
Author Response
We are grateful for your expert feedback, which has significantly strengthened our study. The manuscript has been rigorously revised to reflect your suggestions, resulting in meaningful enhancements. Below, we provide our point-by-point response (in regular font) to your original comments (in bold and italics). The quotations from the manuscript is marked as italics and red color.
- Thank you for the opportunity of reading the results of your research. The topic of the study is interesting and relevant for the actual economic and educational concerns providing solid practical inputs to a better employee selection.
Thank you for your kind words about our research. We sincerely appreciate you taking the time to review our work and for affirming its potential contributions to both academic and practical domains of employee selection.
Your positive feedback is truly encouraging as we continue to explore this important area.
- The theoretical background of the study is well references but could be improved by approaching some social theories such as Self-Efficacy Theory, Self-determination theory, Multiple Intelligence Theory of other learning/Social theories. It is important to see if the academical environment and final evaluation are truly the main factors that contributed to the results of 500 Forbes members and not other factors and how social and learning theories explain the results.
Thank you for your valuable suggestion to strengthen the theoretical foundation of our study. We have revised the Introduction to incorporate a dedicated theoretical framework section (lines 29-51), as follows:
‘In the highly competitive job market, resume screening serves as the first critical checkpoint in corporate recruitment, where its efficiency and criteria directly impact the effectiveness and fairness of human resource management (Ling et al., 2024). As the initial step in the hiring process, resumes not only function as a “stepping stone” for job seekers to showcase their strengths and secure interview opportunities but also represent a crucial mechanism for employers to efficiently identify talent and enhance the equity and efficacy of recruitment practices. This process aligns with Dual-Process Models of Impression Formation (Bellucci, 2023; Freeman & Ambady, 2011), which posit that evaluators rely on both rapid, heuristic-based judgments and deliberate, systematic processing when assessing candidates.
The prominence of educational prestige in resume screening exemplifies the heuristic pathway of impression formation. Graduates from elite universities enjoy distinct advantages in recruitment processes (Moore et al., 2023; Pinto et al., 2024; Ramamurthy & Sedgley, 2019), a phenomenon rooted in recruiters’automatic associations between institutional reputation and perceived competence (Campbell et al., 2019; Pinto et al., 2024). There exists neurocognitive evidence showing how status markers activate neural reward circuits (Bowyer et al., 2021) and reduce cognitive conflict signals in evaluative tasks (Jia et al., 2021). Conversely, the critical role of internship experiences reflects the systematic processing pathway, where recruiters engage in effortful analysis of candidates’practical skills and industry exposure (Li & Lu, 2023; Benati et al., 2023). These dual processes create inherent path in resume evaluation, while elite credentials facilitate cognitive efficiency through schema-driven judgments, high-quality internships demand resource-intensive verification of skill-specific evidence.’
Reference
Bellucci, G. (2023). The organizational principles of impression formation. Cognition, 239, 105550. https://doi.org/10.1016/j.cognition.2023.105550
Freeman, J. B., & Ambady, N. (2011). A dynamic interactive theory of person construal. Psychological Review, 118(2), 247–279. https://doi.org/10.1037/a0022327
Bowyer, C., Brush, C. J., Threadgill, H., Harmon-Jones, E., Treadway, M., Patrick, C. J., & Hajcak, G. (2021). The effort-doors task: Examining the temporal dynamics of effort-based reward processing using ERPs. NeuroImage, 228, 117656. https://doi.org/10.1016/j.neuroimage.2020.117656
Jia, Y., Cui, L., Pollmann, S., & Wei, P. (2021). The interactive effects of reward expectation and emotional interference on cognitive conflict control: An ERP study. Physiology & Behavior, 234, 113369. https://doi.org/10.1016/j.physbeh.2021.113369
And we also use Dual-Process Models of Impression Formation to explain the results, as detailed from lines 318 to 327:
‘The P200 reflects an attentional orienting phase where recruiters allocate selective attention based on academic background, as evidenced by differential activation in the temporo-parietal junction. The subsequent semantic integration stage indexed by N300 reveals how prefrontal-striatal circuits evaluate congruence between educational institutions and internship quality and ultimately influences evaluation outcomes. These two stages provide a novel perspective for understanding cognitive biases in resume screening, emphasizing that educational background not only shapes early attention but also influences later evaluation processes related to self-concept consistency. This temporal hierarchical structure provides neurophysiological evidence for Dual-Process Models of Impression Formation (Amodio, 2019; Monroe et al., 2018).’
Reference
Amodio, D. M. (2019). Social Cognition 2.0: An Interactive Memory Systems Account. Trends in Cognitive Sciences, 23(1), 21–33. https://doi.org/10.1016/j.tics.2018.10.002
Monroe, B. M., Koenig, B. L., Wan, K. S., Laine, T., Gupta, S., & Ortony, A. (2018). Re-examining dominance of categories in im-pression formation: A test of dual-process models. Journal of Personality and Social Psychology, 115(1), 1–30. https://doi.org/10.1037/pspa0000119
- Statistical methods should be more clear connected with the theoretical background, better hypotheses formulation would help the reader understand what the results represent according to previous theory and statistics research on the topic.
Thank you for this valuable suggestion. We have carefully revised the manuscript to improve the clarity and traceability of our hypotheses. Specifically:
- In the Introduction, we now present the hypotheses as a numbered list (H1, H2) in a dedicated paragraph before the Methods section (lines 88-94). Each hypothesis explicitly references the predicted neural markers (P200 and N300) and their cognitive interpretations.
‘Based on these neurocognitive mechanisms, we propose the following hypotheses:
H1: Elite university candidates with Fortune 500 internships will elicit significantly enhanced P200 amplitudes compared with other conditions, reflecting prioritized allocation of early attentional resources.
H2: Elite university candidates with Fortune 500 internships (high semantic coherence condition) will elicit significantly attenuated N300 amplitudes compared to those with elite university and non-Fortune 500 internship (low semantic coherence condition).’
- In the Results, we clearly explain whether the hypothesis has been verified, as detailed on 6 and p.7, lines 226-234 and lines 244-254.
‘As hypothesized in H1, ... According to the results above, H1 got supported.’ and ‘Consistent with H2, ... According to the results above, H2 got supported.’
- The results should compare more studies of the impact of the stereotypical cognitive pattern termed the “elite university effect” on the success of top fortunes. This argument is not very well sustained by the study discussions and conclusions.
- Thank you for your valuable suggestion to strengthen the comparative analysis of our findings. We have revised the Discussion section to include:
‘In the present study, the P200 component was successfully elicited with enhanced amplitude over central regions, aligning with prior ERP studies implicating the central-parietal network in early sensory processing and stimulus salience detection (Irak et al., 2019), reflecting heightened sensory processing in central regions when participants evaluate career-related information.’(see lines 280-284)
‘Notably, internship information from Fortune 500 corporations elicited significantly greater P200 amplitudes compared to non-top-tier enterprises, this finding corroborated behavioral economics research demonstrating that prestigious brands or employers automatically capture attention due to their association with social rewards (Tang & Gray, 2021),’(see lines 284-288)
‘This aligned with the “expectancy-value” framework in cognitive neuroscience, where neural response to incentives depend on both perceived value and subjective expectancy of goal attainment (Zeineldin et al., 2022).’(see lines 294-296)
‘The current investigation revealed enhanced N300 amplitudes over frontal electrode clusters, aligned with the established role of frontal cortices in conflict monitoring and higher-order cognitive control (Fraga et al., 2018),’(see lines 301-303)
‘Conversely, non-Fortune 500 internship records elicited heightened N300 responses in elite academic contexts, this indicated that the combination of non-Fortune 500 companies and elite academic backgrounds violated the expectations,’(see lines 307-309)
Reference
Irak, M., Soylu, C., Turan, G., & Çapan, D. (2019). Neurobiological basis of feeling of knowing in episodic memory. Cognitive Neu-rodynamics, 13(3), 239–256. https://doi.org/10.1007/s11571-019-09520-5
Tang, S., & Gray, K. (2021). Feeling empathy for organizations: Moral consequences, mechanisms, and the power of framing. Jour-nal of Experimental Social Psychology, 96, 104147. https://doi.org/10.1016/j.jesp.2021.104147
Zeineldin, M., Patel, A. G., & Dyer, M. A. (2022). Neuroblastoma: When differentiation goes awry. Neuron, 110(18), 2916–2928. https://doi.org/10.1016/j.neuron.2022.07.012
Fraga, F. J., Mamani, G. Q., Johns, E., Tavares, G., Falk, T. H., & Phillips, N. A. (2018). Early diagnosis of mild cognitive impair-ment and Alzheimer’s with event-related potentials and event-related desynchronization in N-back working memory tasks. Computer Methods and Programs in Biomedicine, 164, 1–13. https://doi.org/10.1016/j.cmpb.2018.06.011
- We still thoroughly revised the Conclusion section to better link the experimental results, as detailed from lines 345-355:
‘The study delineates a neurocognitive trajectory through which graduate institution prestige and internship enterprise status modulate distinct stages of career-related information processing. The P200 and N300 findings collectively reveal a spatiotemporal dynamic: high-prestige stimuli (Fortune 500 enterprises) first capture amplified attentional resources in central regions (indexed by P200), followed by reduced frontal conflict signals (indexed by N300) specifically in elite university graduates. This two-stage mechanism, early salience prioritization and later schema-driven conflict resolution, suggests that institutional identity hierarchically gates neural responses, with elite graduates exhibiting both enhanced motivated attention to prestigious opportunities and diminished cognitive dissonance when integrating these stimuli into self-relevant frameworks.’
- Practical contributions are very clear stated, but the study must underline a more solid scientific and theoretical background, also connect the obtained results with previous similar results.
- Thank you for your valuable suggestion to strengthen the theoretical foundation of our study. We have revised the Introduction to incorporate a dedicated theoretical framework section (lines 29-51), as follows:
‘In the highly competitive job market, resume screening serves as the first critical checkpoint in corporate recruitment, where its efficiency and criteria directly impact the effectiveness and fairness of human resource management (Ling et al., 2024). As the initial step in the hiring process, resumes not only function as a “stepping stone” for job seekers to showcase their strengths and secure interview opportunities but also represent a crucial mechanism for employers to efficiently identify talent and enhance the equity and efficacy of recruitment practices. This process aligns with Dual-Process Models of Impression Formation (Bellucci, 2023; Freeman & Ambady, 2011), which posit that evaluators rely on both rapid, heuristic-based judgments and deliberate, systematic processing when assessing candidates.
The prominence of educational prestige in resume screening exemplifies the heuristic pathway of impression formation. Graduates from elite universities enjoy distinct advantages in recruitment processes (Moore et al., 2023; Pinto et al., 2024; Ramamurthy & Sedgley, 2019), a phenomenon rooted in recruiters’ automatic associations between institutional reputation and perceived competence (Campbell et al., 2019; Pinto et al., 2024). There exists neurocognitive evidence showing how status markers activate neural reward circuits (Bowyer et al., 2021) and reduce cognitive conflict signals in evaluative tasks (Jia et al., 2021). Conversely, the critical role of internship experiences reflects the systematic processing pathway, where recruiters engage in effortful analysis of candidates’practical skills and industry exposure (Li & Lu, 2023; Benati et al., 2023). These dual processes create inherent path in resume evaluation, while elite credentials facilitate cognitive efficiency through schema-driven judgments, high-quality internships demand resource-intensive verification of skill-specific evidence.’
Reference
Bellucci, G. (2023). The organizational principles of impression formation. Cognition, 239, 105550. https://doi.org/10.1016/j.cognition.2023.105550
Freeman, J. B., & Ambady, N. (2011). A dynamic interactive theory of person construal. Psychological Review, 118(2), 247–279. https://doi.org/10.1037/a0022327
Bowyer, C., Brush, C. J., Threadgill, H., Harmon-Jones, E., Treadway, M., Patrick, C. J., & Hajcak, G. (2021). The effort-doors task: Examining the temporal dynamics of effort-based reward processing using ERPs. NeuroImage, 228, 117656. https://doi.org/10.1016/j.neuroimage.2020.117656
Jia, Y., Cui, L., Pollmann, S., & Wei, P. (2021). The interactive effects of reward expectation and emotional interference on cognitive conflict control: An ERP study. Physiology & Behavior, 234, 113369. https://doi.org/10.1016/j.physbeh.2021.113369
- Thank you for your valuable suggestion to strengthen the comparative analysis of our findings. We have revised the Discussion section to include:
‘In the present study, the P200 component was successfully elicited with enhanced amplitude over central regions, aligning with prior ERP studies implicating the central-parietal network in early sensory processing and stimulus salience detection (Irak et al., 2019), reflecting heightened sensory processing in central regions when participants evaluate career-related information.’(see lines 280-284)
‘Notably, internship information from Fortune 500 corporations elicited significantly greater P200 amplitudes compared to non-top-tier enterprises, this finding corroborated behavioral economics research demonstrating that prestigious brands or employers automatically capture attention due to their association with social rewards (Tang & Gray, 2021),’(see lines 284-288)
‘This aligned with the “expectancy value” framework in cognitive neuroscience, where neural response to incentives depend on both perceived value and subjective expectancy of goal attainment (Zeineldin et al., 2022).’(see lines 294-296)
‘The current investigation revealed enhanced N300 amplitudes over frontal electrode clusters, aligned with the established role of frontal cortices in conflict monitoring and higher-order cognitive control (Fraga et al., 2018),’(see lines 301-303)
‘Conversely, non-Fortune 500 internship records elicited heightened N300 responses in elite academic contexts, this indicated that the combination of non-Fortune 500 companies and elite academic backgrounds violated the expectations,’(see lines 307-309)
Reference
Irak, M., Soylu, C., Turan, G., & Çapan, D. (2019). Neurobiological basis of feeling of knowing in episodic memory. Cognitive Neu-rodynamics, 13(3), 239–256. https://doi.org/10.1007/s11571-019-09520-5
Tang, S., & Gray, K. (2021). Feeling empathy for organizations: Moral consequences, mechanisms, and the power of framing. Jour-nal of Experimental Social Psychology, 96, 104147. https://doi.org/10.1016/j.jesp.2021.104147
Zeineldin, M., Patel, A. G., & Dyer, M. A. (2022). Neuroblastoma: When differentiation goes awry. Neuron, 110(18), 2916–2928. https://doi.org/10.1016/j.neuron.2022.07.012
Fraga, F. J., Mamani, G. Q., Johns, E., Tavares, G., Falk, T. H., & Phillips, N. A. (2018). Early diagnosis of mild cognitive impair-ment and Alzheimer’s with event-related potentials and event-related desynchronization in N-back working memory tasks. Computer Methods and Programs in Biomedicine, 164, 1–13. https://doi.org/10.1016/j.cmpb.2018.06.011
Round 2
Reviewer 1 Report
Comments and Suggestions for Authors
Please, refine the title for grammatical accuracy and make the abstract more concise while clearly stating the sample size.
Methodological descriptions should be expanded: explain how trials were randomised, report post-artefact ERP trial counts and rejection rates, and specify any behavioural exclusion criteria. In reporting statistics, give exact p-values and confidence intervals, and treat marginal effects consistently.
The discussion overstates causality, tone down claims about the “halo synergy” mechanism or provide mediation analyses, and recognise alternative explanations such as familiarity effects. Because the sample consists solely of HR students, emphasise the limited generalisability and recommend replication with professional recruiters.
Comments on the Quality of English LanguageShorten long sentences, correct minor grammatical errors and ensure figures are more self-contained with clearer labels.
Author Response
Thank you very much for your helpful suggestions. Our paper has benefited substantially from your insightful comments, resulting in a much-improved paper in many respects. In the following, we provide our point-by-point responses (in regular font) to your original comments (reproduced in bold and italics). The quotations from the manuscript is marked as italics and red color.
- Please, refine the title for grammatical accuracy and make the abstract more concise while clearly stating the sample size.
We sincerely appreciate your constructive comments. We have carefully addressed all your suggestions as follows:
- Title refinement:
‘How Educational Background Influences Recruitment Evaluation: Evidence from Event-Related Potentials’(see lines 1-3)
- Abstract improvements:
‘This study used event-related potentials (ERPs) to examine how candidates’ educational background (elite vs. non-elite universities) and prior internship experience (Fortune 500 vs. non-Fortune 500 enterprises) influence recruitment evaluations. Thirty-two participants completed a 2×2 within-subjects design task. Behavioral data indicated that applicants with Fortune 500 internships and graduates from elite university received higher evaluation scores. ERP results revealed that Fortune 500 experience elicited larger P200 amplitudes (reflecting early attention). Crucially, this effect was modulated by educational background, only candidates from elite universities showed both enhanced P200 and reduced N300 amplitudes (suggesting efficient later processing). These findings indicate that recruiters dynamically allocate attention based on academic prestige (P200) and evaluate semantic congruence between education and employer reputation (N300), providing neurophysiological evidence for educational bias in hiring.’(see lines 7-17)
- Methodological descriptions should be expanded: explain how trials were randomised, report post-artefact ERP trial counts and rejection rates, and specify any behavioural exclusion criteria. In reporting statistics, give exact p-values and confidence intervals, and treat marginal effects consistently.
We sincerely appreciate your thorough review and valuable methodological suggestions. We have carefully addressed each point in the revised manuscript as follows:
- The study consisted of 6 fixed-order blocks (as determined by our paradigm design requirements). Within each block, all 32 trials were presented in fully randomized order, as detailed on page 4 lines 146-147:
‘The experimental structure comprised 6 blocks, each containing 32 trials (total N = 192 trials), presented in fully randomized order.’
- After artifact rejection, an average of 44.7 (SD = 4.5) trials per condition remained. Trials exceeding ±100 μV or containing eye blinks were excluded, with an average rejection rate of 7.4%, as detailed on page 4 lines 167-169:
‘After artifact rejection, an average of 44.7 (SD = 4.5) trials per condition remained. Trials exceeding ±100μV or containing eye blinks were excluded, with an average rejection rate of 7.4%.’
- Our participant exclusion criteria were guided by Wessel’s standards. Results showed that all recruited participants successfully met these predefined criteria. Participants with more than 15% of trials with response time below 100 ms were excluded from analysis, as detailed on page 4 lines 169-171:
‘Additionally, participants were excluded from analysis if more than 15% of their trials had a response time below 100 ms (Wessel, 2018). All recruited participants successfully met this predefined criterion.’
Reference
Wessel, J. R. (2018). An adaptive orienting theory of error processing. Psychophysiology, 55(3), e13041.
- Statistical Reporting is revised, as detailed on page 5-6:
‘A repeated measures ANOVA with factors of graduate institution (elite vs. non-elite) and internship enterprise (Fortune 500 vs. Non-Fortune 500) was conducted on the job application ratings. The analysis revealed a significant main effect of internship enterprise, indicating that job applicants with Fortune 500 internship experience received significantly higher ratings, (F(1, 31) = 52.04, p = .0003, η²p = .63, 95%CI [4.739, 15.823]). A significant main effect of graduate institution was also found, graduates of elite universities have obvious advantages in scoring, (F(1, 31) = 120.46, p = .0002, η²p = .80, 95%CI [9.383, 22.612]). Importantly, the interaction between graduate institution and internship enterprises was marginal significant (F(1, 31) = 3.60, p = .070, η²p = .09, 95%CI [-0.038, 7.383]). Although the effect approached significance (p = .070), it did not reach the conventional threshold of .05.’
‘A 2 (graduate institution: elite vs. non-elite) × 2 (internship enterprise: Fortune 500 vs. non-Fortune 500) × 2 (brain region: frontal vs. central) repeated-measures ANOVA conducted on electrodes spanning frontal (F3, Fz, F4) and central (C3, Cz, C4) regions revealed significant findings. The main effect of brain region was significant, with greater P200 amplitudes observed in central regions (M = 1.00, SE = 0.39) compared to frontal regions (M = 0.40, SE = 0.43), F(1, 31) = 8.84, p = .006, η²p = .22, 95%CI [1.170, 8.351]. The main effect of internship enterprise reached significance, demonstrating larger P200 amplitudes elicited by Fortune 500 corporate information (M = 1.19, SE = 0.43) relative to non-Fortune 500 enterprises (M = 0.21, SE = 0.42), F(1, 31) = 12.71, p = .001, η²p = .29, 95%CI [3.181, 9.873]. As hypothesized in H1, a significant interaction emerged between graduate institution and internship enterprise, F(1, 31) = 4.95, p = .033, η²p = .14, 95%CI [0.463, 2.048] (see Figure 3). Simple effects analysis revealed that among graduates from elite universities, Fortune 500 internship conditions elicited significantly enhanced P200 amplitudes (M = 2.61, SE = 0.53) compared to non-Fortune 500 conditions (M = 0.95, SE = 0.54), F(1, 31) = 15.30, p = .002, η²p = .33, 95%CI [0.574, 2.809]. No significant differences were observed in the non-elite university group (p = .434). The mean P200 amplitude was detailed in Figure 5, with corresponding topographic mappings presented in Figure 6. According to these results, H1 was supported.’
‘A 2 (graduate institution: elite vs. non-elite) × 2 (internship enterprise: Fortune 500 vs. non-Fortune 500) × 2 (brain region: frontal vs. central) repeated-measures ANOVA conducted on electrodes spanning frontal (F3, Fz, F4) and central (C3, Cz, C4) regions revealed significant findings. The main effect of brain region was significant, with greater N300 amplitudes observed in frontal regions (M = -3.24, SE = 0.48) compared to central regions (M = -1.82, SE = 0.49), F(1, 31) = 66.66, p = .0006, η²p = .68, 95%CI [-4.838, -0.238]. Consistent with H2, the interaction between graduate institution and internship enterprise was significant for N300 amplitude, F(1, 31) = 9.77, p = .004, η²p = .24, 95%CI [-6.583, -0.428] (see Figure 4.). Simple effects analysis revealed that among graduates from elite universities, Fortune 500 internship conditions elicited significantly reduced N300 amplitudes (M = -2.98, SE = 0.61) compared to non-Fortune 500 conditions (M = -4.65, SE = 0.57), F(1, 31) = 4.30, p = .006, η²p = .12, 95%CI [-6.196, -0.265]. No significant differences were observed in the non-elite university group (p = .163). Neither the main effect of graduate institution nor internship organization, nor other interaction effects reached statistical significance (p = .571). The mean N300 amplitude was detailed in Figure 5, with corresponding topographic mappings presented in Figure 6. According to these results, H2 was supported.’
- The discussion overstates causality, tone down claims about the “halo synergy” mechanism or provide mediation analyses, and recognise alternative explanations such as familiarity effects. Because the sample consists solely of HR students, emphasise the limited generalisability and recommend replication with professional recruiters.
Thank you for this insightful suggestion. Your comments are helpful for us to enhance the scientific rigor, clarity and readability of the manuscript. Below, we address each point raised:
- We fully agree that the causal implications of the “halo synergy” mechanism in the original discussion exceeded the correlational nature of our experimental design. We have revised the language to temper causal claims (e.g., changing “halo synergy” to “association”). Revision about toning down claims about the “halo synergy” mechanism, as detailed on page 9 lines 300-304 :
‘This temporal dynamic pattern provides tentative support for the hypothesized dissociation and interaction between automatic heuristic processing and controlled evaluation in personnel selection. Notably, we observed a neural-level association pattern between elite degrees and Fortune 500 internship experiences. This finding contributes to understanding how cognitive schemas could shape social evaluations.’
- Your suggestion regarding the consideration of alternative explanations, especially the familiarity effect, is extremely timely and important. We acknowledge that due to the within-subjects experimental design employed in this study, each participant was exposed to all experimental conditions or materials. This design is more susceptible to familiarity effect, where participants may become increasingly familiar with the experimental tasks, materials, or measurement procedures over time, potentially influencing subsequent performance and providing alternative explanations for the observed results. During the initial stages of experimental design, we carefully considered this issue and took proactive measures to mitigate familiarity effects. Specifically, we incorporated a larger variety and quantity of experimental materials and randomized the presentation order of stimuli to minimize the impact of familiarity effect. We have added a new point in the "Limitations and Future Directions" section, clearly acknowledging and discussing the familiarity effect, as detailed on page 10 lines 334-338:
‘Third, while our standardized stimuli controlled for extraneous variables, the repetitive presentation of stylized candidate profiles introduces potential familiarity effects. Such learned schematic processing may systematically bias participants’ responses. Future research should consider employing novel or less conventional stimuli.’
- We fully agree with your concerns regarding the composition of the sample (limited to HR students) and its impact on the generalizability of the research results. This is a very crucial limitation. We have significantly strengthened the emphasis on this limitation in the "Limitations and Future Directions" section, as detailed on page 10 lines 328-334:
‘Second, as our sample consisted exclusively of HR management students (albeit with internship experience), the findings have inherent limitations in generalizability to actual recruitment contexts. We therefore strongly recommend that future research prioritize replication with practicing HR professionals as the primary validation step. Subsequent extensions could then examine: (a) expert-novice comparisons in evaluator cognition, (b) cross-cultural cohorts to assess sociocultural modulation effects, and (c) age-diverse populations to explore developmental aspects of bias formation.’
- Shorten long sentences, correct minor grammatical errors and ensure figures are more self-contained with clearer labels.
We appreciate your valuable feedback. We have carefully addressed your suggestions as follows:
- We have shortened lengthy sentences for clarity and corrected minor grammatical errors throughout the manuscript.
- We have enhanced self-containment by adding clear labels and improved axis descriptions and units for better interpretability, as detailed on page 6-7:
‘Figure 2. The results of P200 amplitude (μV) in each condition. Error bars represent standard errors (n.s.: nonsignificant, *p < 0.05, **p < 0.01,***p < 0.001).’
‘Figure 3. The results of N300 amplitude (μV) in each condition. Error bars represent standard errors(n.s.: nonsignificant, *p < 0.05, **p < 0.01,***p < 0.001).’
‘Figure 4.ERPs Results. The grand average waveforms at six representative electrodes in the midzone (Fz, F3, F4, Cz, C3, and C4) are presented. Positive voltage is plotted downward. The x-axis represents time (ms) relative to stimulus onset (0 ms), and the y-axis represents amplitude values (μV).’
‘Figure 5. P200 (Cz, 210-280ms) and N300 (Fz, 300-350ms) scalp distributions across elite/non-elite × Fortune/non-Fortune 500 conditions.’
